# Radiation Exposure among Patients with Inflammatory Bowel Disease: A Single-Medical-Center Retrospective Analysis in Taiwan

**DOI:** 10.3390/jcm11175050

**Published:** 2022-08-28

**Authors:** Chen-Ta Yang, Hsu-Heng Yen, Yang-Yuan Chen, Pei-Yuan Su, Siou-Ping Huang

**Affiliations:** 1Division of Gastroenterology, Changhua Christian Hospital, Changhua 500, Taiwan; 2College of Medicine, National Chung Hsing University, Taichung 400, Taiwan; 3Department of Electrical Engineering, Chung Yuan Christian University, Taoyuan 320, Taiwan; 4General Education Center, Chienkuo Technology University, Changhua 500, Taiwan; 5Department of Hospitality Management, MingDao University, Changhua 500, Taiwan

**Keywords:** inflammatory bowel disease, Crohn’s disease, radiation, imaging

## Abstract

Inflammatory bowel disease (IBD) is a chronic and relapsing disease that can be complicated by abscesses, fistulas, or strictures of the damaged bowel. Endoscopy or imaging studies are required to diagnose and monitor the treatment response or complications of the disease. Due to the low incidence of the disease in Taiwan, the pattern of radiation exposure from medical imaging has not been well studied previously. This retrospective study aimed to evaluate the pattern of radiation exposure in 134 Taiwanese IBD patients (45 CD and 89 UC) diagnosed and followed at Changhua Christian Hospital from January 2010 to December 2020. We reviewed the patient demographic data and radiation-containing image studies performed during the follow-up. The cumulative effective dose (CED) was calculated for each patient. During a median follow-up of 4 years, the median CED was higher for patients with CD (median CED 21.2, IQR 12.1–32.8) compared to patients with UC (median CED 2.1, IQR 0–5.6) (*p* < 0.001). In addition, the CD patients had a trend of a higher rate of cumulative ≥50 mSv compared with the UC patients (6.7% vs. 1.1%, *p* = 0.110). In conclusion, our study found a higher radiation exposure among CD patients compared to patients with UC, representing the complicated nature of the disease. Therefore, increasing the use of radiation-free medical imaging such as intestinal ultrasound or magnetic resonance imaging should be advocated in daily practice to decrease the risk of excessive radiation exposure in these patients.

## 1. Introduction

Inflammatory bowel disease (IBD), including ulcerative colitis (UC) and Crohn’s disease (CD), is chronic, fluctuating, and relapsing. Due to the complexity of the disease and complications such as strictures, fistulas, or perforations, medical image studies such as computed tomography are usually required to diagnose the disease and manage the complications [1,2,3].

Due to its high accuracy and accessibility, computed tomography, one of the most commonly utilized medical imaging tools, has seen a marked increase over the past decades. However, there is a concern regarding subsequent cancer development due to repetitive medical radiation exposure [4]. For example, an abdominal computed tomography (CT) scan provides an effective radiation dose of 10 mSv, which is 4.2-fold of the background radiation per year worldwide (2.4 mSv). In addition, due to IBD’s chronic and complicated nature, endoscopic and radiological examinations are frequently required to monitor the disease course to reach the therapeutic goals for “Treat-to-Target” strategies in IBD [5].

A meta-analysis that included six studies of 1704 participants showed that 8.8% of IBD patients 11.1% of Crohn’s disease patients, and 2% of ulcerative colitis patients received potentially harmful levels of radiation (defined as ≥50 mSv) [6]. Nguyen et al. [7] reported a population-based cohort of IBD patients from Canada between 1994 and 2016 and found Crohn’s disease patients were more likely than UC patients to have excessive radiation exposure (15.6% vs. 6.2%; *p* < 0.001). Although these previous studies show an increased risk of cumulated radiation exposure among IBD patients [7,8,9,10,11], no previous report addressed this issue among the Taiwanese population. As the incidence and prevalence of IBD are increasing in Taiwan [12,13,14,15,16,17,18], it is essential to understand the pattern of radiation exposure in this growing population. Therefore, we conducted this study to estimate the radiation exposure and analyze possible risk factors associated with radiation exposure among Taiwanese IBD patients.

## 2. Materials and Methods

### 2.1. Study Design and Patients

This retrospective study was conducted in a tertiary medical center in central Taiwan. The inclusion criteria included: (a) a diagnosis of either Crohn’s disease or ulcerative colitis certified by the Ministry of Health and Welfare of Taiwan; (b) patients aged ≥18 years at diagnosis; (c) diagnosis made after January 2010 in our hospital; (d) no previous diagnosis of malignancy before the diagnosis of IBD; and (e) follow-up for at least three months in the clinic for disease treatment. From January 2010 to December 2020, 163 adult patients with a diagnosis of inflammatory bowel disease were identified and 137 were diagnosed after January 2010 in our hospital. Three of them were excluded due to a diagnosis of malignancy made before the IBD diagnosis. The duration of follow-up was calculated from the first and last outpatient visits during the study period. In addition, the demographic, clinical (including the extent of involved GI tract and modality of treatment), and radiological data were collected. The study was approved by the Institutional Review Board (IRB) of Changhua Christian Hospital (IRB number: CCH IRB No 210202, group 4 of the study population).

### 2.2. Medical Radiation Exposure

A gray (Gy) represents an absorbed dose of radiation that is the deposit of a joule of radiation energy per kg of tissue. A Sievert (Sv) represents the equivalent biological effect (effective dose) of radiation and is the absorbed dose’s product and a radiation weighting factor. Because the radiation weighting factor is 1.0 for X-rays and gamma rays, 1 Gy is equivalent to 1 Sv in medical imaging. The radiation dose in medical imaging is typically expressed as millisieverts (mSv). The type and number of each radiological image were counted and the cumulative effective dose (CED) was calculated for each patient. The mean radiation dose for each standard radiological imaging is listed in Table 1 [10].

### 2.3. Statistical Analysis

Data are expressed as *n* (%), mean ± standard deviation (SD), or median (interquartile range (IQR)), as the distribution of continuous variables was normal or non-normal as per the one-sample Kolmogorov–Smirnov test. We compared the differences in the descriptive characteristics of the study participants between 87 Crohn’s disease and ulcerative colitis patients using a chi-squared test, a Fisher’s exact test (categorical 88 data), the Student’s *t*-test, or the Mann–Whitney U-test (continuous data), as appropriate. A Kaplan–Meier analysis showed the difference in the cumulative probability of exposure to CED ≥ 50 mSv after diagnosis between inflammatory bowel disease (IBD) types. All data were analyzed using PS IMAGO Pro 7 software (IBM, Armonk, NY, USA); *p* < 0.05 indicated statistical significance.

## 3. Results

Among 137 adult patients, 3 were excluded due to a cancer diagnosis before IBD. The remaining 134 subjects entered the final analysis, including 45 subjects with Crohn’s disease and 89 with ulcerative colitis. The flowchart for the study is shown in Figure 1.

Table 2 shows the clinical features of the study population. There were more male patients (61.2%). There was no difference in patients receiving 5-ASA or steroid treatment between CD and UC. However, a higher proportion of CD patients compared with UC patients received therapy with azathioprine (75.6% vs. 18.0%, *p* < 0.001) or biologics (66.7% vs. 18.0%, *p* < 0.0.001). The CD patients had a high proportion of IBD-related surgery or admission compared with the UC patients. A total of 109 CT scans and 31 magnetic resonance imaging (MRI) scans were performed during the study period. During a median follow-up duration of 4 years, the median CED was 4.9 (IQR 0.7–18.4) for the study population. The median CED was higher for patients with CD (median CED 21.2, IQR 12.1–32.8) compared to patients with UC (median CED 2.1, IQR 0–5.6) (*p* < 0.001). The percentage of CED ≥ 50 mSv was numerically higher among patients with CD than with UC (6.7% vs. 1.1%, *p* = 0.110).

The disease behavior of the study population is shown in Table 3. In 37.1% of our UC patients, extensive colitis was found, followed by left-side colitis (44.9%) and proctitis (18%). Our CD patients presented the penetration type (33.3%) and structuring type (31.1%). The proportion of patients exposed to CED ≥ 50 mSv was higher among patients with CD than those with UC and IBD-related surgery or admissions, as shown in Figure 2.

## 4. Discussion

Our study was the first to investigate the pattern of radiation exposure among Taiwanese IBD patients, and the findings were consistent with reports from the Western population [3,7]. Patients with CD are more complicated than those with UC and have higher biological or immunosuppressive therapy rates. Our study found a higher radiation exposure among these patients, highlighting the complicated nature of the disease in Taiwan.

The growing utilization of CT scans in the United States was rapid from 2000 to 2006 for adults and children, with annual growth rates of 11.6% and 10.1%, respectively. While the use of CT scans still increased in 2013–2016 for adults, the annual growth was much less rapid, with a rate of 3.7% [19]. In contrast to that for adults, the annual growth in CT utilization in 2013–2016 was stabilized among children, with a rate of 0.8%. This data showed growing recognition of radiation exposure among CT scans, especially in the pediatric population.

A cross-section image is important in diagnosing IBD, managing complications such as abscesses, and evaluating treatment response. The current consensus on therapeutic goals to achieve the “Treat-to-Target” strategies recommends the clinical response as a short-term target, clinical remission as an intermediate target, and endoscopic healing as a long-term target. A cross-sectional image can help evaluate the target of transmural healing in Crohn’s disease and is regarded as an adjuvant to endoscopic assessment for mucosal healing [5,20,21].

Median radiation exposure among IBD patients ranges from 7.2 to 26.6 mSv for CD patients and from 2.8 to 10.5 mSv for UC patients [10]. The cutoff level for the definition of a high radiation dose differed between existing studies, which defined it as 50, 75, or 100 mSv [8,9,22,23]. The percentage of CD compared with UC might influence the choice of different cutoffs for high radiation doses. In the studies composed mainly of CD patients, 75 mSv or 100 mSv were chosen as the cutoff level for a high radiation dose [10,23]. On the other hand, for the studies including mainly UC patients, 50 mSv was chosen as the cutoff level [24,25]. The duration of IBD also influenced the radiation exposure among this population. Since more UC patients were in our study population and the median follow-up duration was 4 years, we chose 50 mSv as the cutoff level for high radiation doses.

Several studies addressed the risk factors of increased radiation exposure among IBD patients. Most identified previous surgery as a significant risk factor for increased radiation exposure [8,22,23,24,25,26]. In addition, other risk factors such as the use of steroids or biological agents, longer disease duration, hospitalization, and penetrating or structuring disease were significant in some of these studies. Our study also found that patients who received surgery or IBD-related hospital admissions during the follow-up had a higher radiation exposure rate. MRI and ultrasound are good alternative imaging methods that do not use radiation. In addition, MRI offers information for extraluminal manifestations such as fistulas, abscesses, or mesenteric change. A meta-analysis of 33 prospective studies showed high sensitivity and specificity without differences in the diagnosis of IBD between ultrasound, MRI, and CT (sensitivities: 89.7%, 93%, and 84.3%, respectively; specificities: 95.6%, 92.8%, and 95.1%, respectively).

Although CT carries the concern of an increased radiation dose, it has the benefits of a higher availability and accessibility, rapid acquisition time, and lower cost than MRI. A CT scan is preferred over MRI in the acute or emergency setting due to its rapid acquisition time and higher availability. Low-dose CT scans are particularly valuable in such a clinical setting. Low-dose CT using iterative reconstruction can reduce the dose of CT from 34.5% to 73.7% compared to standard-dose CT with acceptable diagnostic accuracy and image quality [27,28,29,30,31]. Current deep learning technology can reduce image noise and improve image quality combined with iterative reconstruction [32,33].

Rapid accumulation of the radiation dose was found in the first year of IBD for the image performed to help diagnose and during the percutaneous drainage era. The cumulative effective dose for a drainage episode was 47.50 mSv [11]. Due to high radiation exposure at the initial diagnosis and during the era of CT-guided drainage, it is crucial to decrease the use of CT in another period. A study by S. M. Govani et al. disclosed no significant findings in one of three CT scans in the emergency department, and only 17% of CT scans revealed complications from CD [34]. Therefore, a risk stratification approach in the emergency department may be considered to avoid the overuse of CT scans. In addition, MRI or ultrasound should be considered for patients with higher cumulative doses to avoid further radiation exposure [35,36,37]. Our study revealed 6.7% of Crohn’s disease patients with CED ≥ 50 mSv and suggested MRI or intestinal ultrasound imaging modality for this group of patients.

There were some limitations to our study. First, we only calculated the patients’ radiological exposure from examinations conducted in our hospital after the disease diagnosis. Although our hospital is the major hospital in Changhua County, the radiation exposure estimation may likely have been underestimated because these patients may have had an unscheduled emergency visit to another hospital. In addition, we could not further analyze risk factors for higher doses of ionizing radiation exposure at different value cutoffs due to a small number of cases. Second, the study was conducted in a tertiary medical center. Patients at such facilities are more complicated and the pattern of radiation exposure may not reflect all the patients in the general clinic. Third, we used the same radiation dose for a typical radiological imaging examination (Table 1). The radiation dose could be varied by body weight, gender, age at exposure, or the brand of machine utilized [38], and such estimation may have been inaccurate. Fourth, due to the dynamic course of the disease, we did not analyze disease activity markers such as clinical, endoscopic, laboratory, or disease severity assessment and correlation to imaging at each time point to explore their associations. Fourth, the follow-up period was limited; we could not investigate the potential harm of radiation exposure in our population.

## 5. Conclusions

In conclusion, our study showed that Crohn’s disease was associated with a higher radiation exposure than UC. Furthermore, as the disease is long-lasting, the expected cumulated radiation exposure increased over time. Therefore, clinicians should be encouraged to use a low-radiation-dose protocol for computed tomography or non-ionizing methods such as intestinal ultrasound in their daily practices to decrease the risk of excessive radiation exposure in these patients.

## Figures and Tables

**Figure 1 jcm-11-05050-f001:**
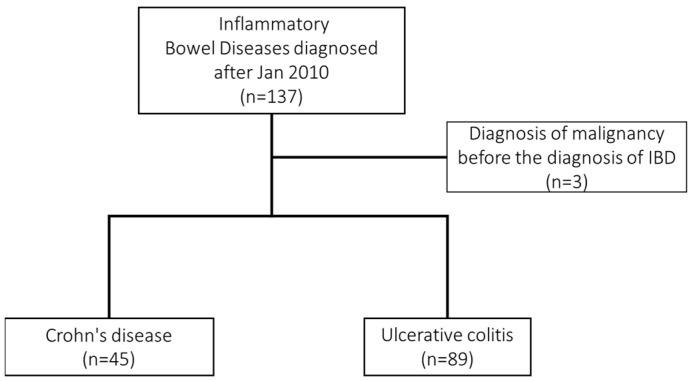
Flow chart of patient enrollment in the present study.

**Figure 2 jcm-11-05050-f002:**
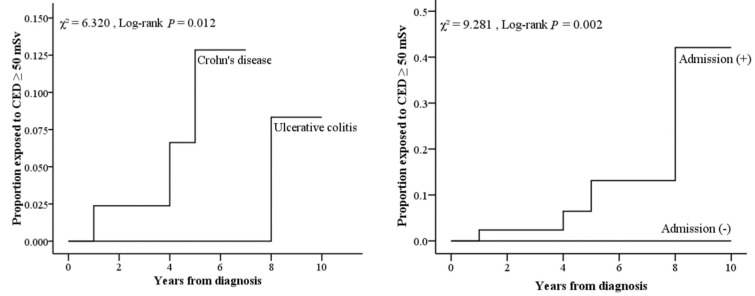
Kaplan–Meier analysis showing a significant difference in the cumulative probability of exposure to CED ≥ 50 mSv after diagnosis between different IBD types, surgery, or hospital admission status.

**Table 1 jcm-11-05050-t001:** The mean radiation dose for each typical radiological imaging type.

Type of Radiological Imaging	Effective Dose (mSv)
Abdominal and pelvic CT	10
Barium enema	8
Upper GI series	6
Small bowel series	5
Abdominal radiography	0.7
Chest radiography	0.02

Abbreviation: CT, computed tomography.

**Table 2 jcm-11-05050-t002:** The baseline characteristics of the study participants.

	IBD (*n* = 134)	Crohn’s Disease	Ulcerative Colitis	*p*-Value
		(*n* = 45)	(*n* = 89)	
Male gender, *n* (%)	82 (61.2%)	28 (62.2%)	54 (60.7%)	0.862
Age at diagnosis, yr, median (IQR)	40 (30–51)	34 (28–48)	43 (33–52)	0.039
Disease duration, yr, median (IQR)	4 (2–5)	4 (3–5)	4 (2–6)	0.891
5-ASA, *n* (%)	126 (94.0%)	40 (88.9%)	86 (96.6%)	0.118
Steroids, *n* (%)	66 (49.3%)	25 (55.6%)	41 (46.1%)	0.299
AZA, *n* (%)	50 (37.3%)	34 (75.6%)	16 (18.0%)	<0.001
Biologics, *n* (%)	46 (34.3%)	30 (66.7%)	16 (18.0%)	<0.001
CED ≥ 50 mSv, *n* (%)	4 (3%)	3 (6.7%)	1 (1.1%)	0.110
Median CED during follow-up, mSv, median (IQR)	4.9 (0.7–18.4)	21.2 (12.1–32.8)	2.1 (0–5.6)	<0.001
Total CT times ≥ 3, *n* (%)	14 (10.4%)	12 (26.7%)	2 (2.2%)	<0.001
Total MRI times ≥ 3, *n* (%)	5 (3.7%)	4 (8.9%)	1 (1.1%)	0.043
Total X-ray, times, median (IQR)	3 (1–7)	8 (4–14)	2 (0–4)	<0.001
Surgery, *n* (%)	15 (11.2%)	14 (31.1%)	1 (1.1%)	<0.0001
IBD-related admission, *n* (%)	46 (34.3%)	28 (62.2%)	18 (20.2%)	<0.0001

Abbreviations: IBD: inflammatory bowel disease; ASA: aspirin; AZA: azathioprine; IQR, interquartile range; CED: cumulative effective dose.

**Table 3 jcm-11-05050-t003:** Location and extent of involvement of IBD.

	Crohn’s Disease	Ulcerative Colitis
	(*n* = 45)	(*n* = 89)
UC Location/disease extent, *n* (%)		
E1: Proctitis	--	16 (18.0%)
E2: Left-side colitis	--	40 (44.9%)
E3: Extensive colitis	--	33 (37.1%)
CD Location/disease extent, *n* (%)		
L1: Ileum	15 (33.3%)	
L2: Colon	5 (11.1%)	
L3: Ileo-colon	23 (51.1%)	
L4: UGI tract	2 (4.4%)	
CD Behaviour/disease behavior, *n* (%)		
B1: Non-stricturing	16 (35.6%)	
B2: Stricturing	14 (31.1%)	
B3: Penetrating	15 (33.3%)	
*p*: Perianal involvement, *n* (%)	4 (8.9%)	

Abbreviations: UC: ulcerative colitis; CD: Crohn’s disease; IBD: inflammatory bowel disease.

## Data Availability

The datasets generated and/or analyzed during the current study are not publicly available, but they may be obtained from the corresponding author upon reasonable request.

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
