# Peer review of "Radiation Exposure among Patients with Inflammatory Bowel Disease: A Single-Medical-Center Retrospective Analysis in Taiwan"

_jcm, 2022, doi:10.3390/jcm11175050_

Round 1

Reviewer 1 Report

This is a small retrospective study in 134 inflammatory bowel disease patients (45 patients with Crohn's disease [CD] and 89 patients with ulcerative colitis [UC]), with a limited follow up of 4 years, showing a higher radiation exposure among CD patients compared to those with UC.

Comments

The only novelty of this study was that it addressed an already researched topic in a Taiwanese population. Previous studies in different populations and a meta-analysis have shown similar results in bigger cohorts and in longer follow ups, the largest one being the one by Nguyen et al in 72,933 incident cases of inflammatory bowel disease in Canada (DOI: 10.1093/ibd/izz219). However, for some reason there still seems to be interest in the literature on this topic, even after 10 years of the publication of the first meta-analysis. The most recent study was published from Poland in December 2021, which was not mentioned by the authors of this study, in a comparable number of patients (65 with CD and 98 with UC), with the same limited follow up of 4 years (Łukawska A, et al. Diagnostics (Basel) 2021 Dec 18;11(12):2387.). Therefore, it would be reasonable to suggest publication of this study from Taiwan after including the most recent literature on the topic such as the one from Poland. In the conclusion, it must be made clear that the novelty of the study was that the cohort was Taiwanese and that the results are comparable to the previous studies from other ethnic populations.

Author Response

Dear Reviewer,

Thank you for reviewing our manuscript and providing your editorial comments as well as reviewers comments for improving our manuscript. Based on these comments, we have made several revisions to our manuscript, which we are hereby resubmitting for your consideration. Our point-by-point responses to the comments are detailed below.

Response to Reviewers’ comments

 The only novelty of this study was that it addressed an already researched topic in a Taiwanese population. Previous studies in different populations and a meta-analysis have shown similar results in bigger cohorts and in longer follow ups, the largest one being the one by Nguyen et al in 72,933 incident cases of inflammatory bowel disease in Canada (DOI: 10.1093/ibd/izz219). However, for some reason there still seems to be interest in the literature on this topic, even after 10 years of the publication of the first meta-analysis. The most recent study was published from Poland in December 2021, which was not mentioned by the authors of this study, in a comparable number of patients (65 with CD and 98 with UC), with the same limited follow up of 4 years (Łukawska A, et al. Diagnostics (Basel) 2021 Dec 18;11(12):2387.). Therefore, it would be reasonable to suggest publication of this study from Taiwan after including the most recent literature on the topic such as the one from Poland. In the conclusion, it must be made clear that the novelty of the study was that the cohort was Taiwanese and that the results are comparable to the previous studies from other ethnic populations.

Response:

  Thank you for your valuable comment. We cite the two references in our revised manuscript. As inflammatory bowel disease has a low prevalence in the Asian population, we believe our present data provide valuable information regarding radiation exposure in the Asian population and are comparable to other ethnic populations.

Thank you for the opportunity to resubmit this manuscript for consideration for publication in the Medicine If you have any questions or comments regarding this manuscript, please do not hesitate to contact us using the details provided below.

Sincerely,

Hsu-Heng Yen, M.D  

Division of Gastroenterology, Department of Internal Medicine, Changhua Christian Hospital, Changhua, Taiwan

Reviewer 2 Report

This study aims to evaluate the pattern of radiation exposure in adult IBD patients showing that Crohn's disease is associated with higher radiation exposure than Ulcerative colitis.  Therefore this study should encourage the use of non-ionizing methods in radiodiastostics in these patients.

Author Response

Dear Reviewer,

Thank you for reviewing our manuscript and providing your editorial comments as well as reviewers comments for improving our manuscript. Based on these comments, we have made several revisions to our manuscript, which we are hereby resubmitting for your consideration. Our point-by-point responses to the comments are detailed below.

Response to Reviewers’ comments

  This study aims to evaluate the pattern of radiation exposure in adult IBD patients showing that Crohn's disease is associated with higher radiation exposure than Ulcerative colitis.  Therefore, this study should encourage the use of non-ionizing methods in radiodiastostics in these patients.

Response:

  Thank you for your valuable comment. We include your suggestion in the revised manuscript.

Thank you for the opportunity to resubmit this manuscript for consideration for publication in the Medicine If you have any questions or comments regarding this manuscript, please do not hesitate to contact us using the details provided below.

Sincerely,

Hsu-Heng Yen, M.D  

Division of Gastroenterology, Department of Internal Medicine, Changhua Christian Hospital, Changhua, Taiwan

Reviewer 3 Report

Radiation exposure in IBD patients is always a matter under consideration especially for Crohn’s disease patients. IUS has gained the interest of the global gastroenterology community and efforts are made to introduce IUS in daily clinical practice to substitute other imaging modalities such as CT or MRE if possible.

This manuscript highlights the increased radiation exposure among IBD patients in Taiwan but there are some points that should be further addressed.

Abstract

·         Be careful with grammar errors and expressions.

Ø  First sentence….. might be complicated with abscess, fistula, or stricture of the damaged bowel.

Ø  2nd sentence :  Endoscopy or imaging studies are required to….

Ø  Rephrase as follows:

This retrospective study aimed to evaluate the pattern of radiation exposure  in 134 IBD Taiwanese patients (45CD, 89UC) diagnosed and followed at Changhua Christian Hospital from January 2010 to December 2020.

Ø  Please omit this sentence:

 We excluded three with a 20 diagnosis of malignancy before the diagnosis of their IBD.

Ø  During a median follow-up duration of 4 years is not correct , please make the necessary changes.

Introduction

Use abbreviations of CD and UC the first time they appear in the manuscript.

 Because of the complexity of the disease and complications such as stricture…..over the past decades. Please include all the relevant references to support that.

Materials & Methods

2.1. Study Design and Patients

Put the total number of patients 137 and then justify that 3 were excluded due to cancer history. Please be also more precise about the inclusion and exclusion criteria, eg inclusion criteria IBD patients aged18years at diagnosis, diagnosis after 19 January 2010, …etc

Any other disease activity markers clinical, endoscopic, laboratory(eg persistent crp elevation)y, or disease severity assessment and correlation to imaging?

Please be more descriptive in this part!

Methodologically you should include more information on the study participants ie demographics, BMI, smoking, number and length of Hospitalizations, emergency department visits(in total) IBD related surgeries, complications such as abscess, toxic megacolon etc, perianal disease, use of more than one biologic agent.

You can also make an analysis(univariate/multivariate) on risk factors for higher doses of ionizing radiation exposure and give a certain(or more than one) cut off for that.

Statistical analysis

The 2nd sentence should be rephrased..

‘Compare the differences in the descriptive characteristics of study participants between 87 Crohn's disease and ulcerative colitis by Chi-square test, the Fisher's exact test (categorical 88 data), the Student’s t-test, or the Mann-Whitney U-test (continuous data), as appropriate.’

Results

Table 3 should be included in table 2.

Based on changes in the method part include relevant results in table 2 and set table 3 for the results of multivariate analysis

  Conclusion

Please correct  cosmography to tomography.

Author Response

Dear Reviewer,

Thank you for reviewing our manuscript and providing your editorial comments as well as reviewers comments for improving our manuscript. Based on these comments, we have made several revisions to our manuscript, which we are hereby resubmitting for your consideration. Our point-by-point responses to the comments are detailed below.

Response to Reviewers’ comments

Radiation exposure in IBD patients is always a matter under consideration especially for Crohn’s disease patients. IUS has gained the interest of the global astroenterology community and efforts are made to introduce IUS in daily clinical practice to substitute other imaging modalities such as CT or MRE if possible.

This manuscript highlights the increased radiation exposure among IBD patients in Taiwan but there are some points that should be further addressed.

#1 Abstract

Be careful with grammar errors and expressions.

Ø  First sentence….. might be complicated with abscess, fistula, or stricture of the damaged bowel.

Ø  2nd sentence :  Endoscopy or imaging studies are required to….

  Rephrase as follows:

This retrospective study aimed to evaluate the pattern of radiation exposure  in 134 IBD Taiwanese patients (45CD, 89UC) diagnosed and followed at Changhua Christian Hospital from January 2010 to December 2020.

Ø  Please omit this sentence:

 We excluded three with a 20 diagnosis of malignancy before the diagnosis of their IBD.

Ø  During a median follow-up duration of 4 years is not correct , please make the necessary changes.

Use abbreviations of CD and UC the first time they appear in the manuscript.

Response:

  Thank you for your valuable suggestion. We made corrections in the revised manuscript.

#2. Introduction

Because of the complexity of the disease and complications such as stricture…..over the past decades. Please include all the relevant references to support that.

Response:

  Thank you for your valuable suggestion. We made the correction in the revised manuscript.

#3 Materials & Methods

2.1. Study Design and Patients

Put the total number of patients 137 and then justify that 3 were excluded due to cancer history. Please be also more precise about the inclusion and exclusion criteria, eg inclusion criteria IBD patients aged≥ 18 years at diagnosis, diagnosis after 19 January 2010, …etc

Response:

  Thank you for your valuable suggestion. We made revisions in the revised manuscript. In the approved IRB protocol, we are only able to access adult patient data (aged≥ 18 years) and we do not have the data of the pediatric population for analysis.

#4. Any other disease activity markers clinical, endoscopic, laboratory(eg persistent crp elevation)y, or disease severity assessment and correlation to imaging? Please be more descriptive in this part!

Response:

  Thank you for your valuable suggestion. As the disease course of IBD is dynamic over time, imaging studies are expected to conduct while the patient has a worsening disease activity. The study aimed to investigate the cumulative dose of radiation exposure and we do not analyze the data at the individual time point of the patient. We agree with the cumulative radiation exposure might be a reflection of disease activity.

#5. Methodologically you should include more information on the study participants ie demographics, BMI, smoking, number, and length of Hospitalizations, emergency department visits(in total) IBD related surgeries, complications such as abscess, toxic megacolon etc, perianal disease, use of more than one biologic agent. You can also make an analysis(univariate/multivariate) on risk factors for higher doses of ionizing radiation exposure and give a certain(or more than one) cut off for that.

Response:

  Thank you for your valuable suggestion. As IBD remains a rare disease in the Taiwanese population, your mentioned events, such as hospitalization, ER visits, or IBD related surgeries is low in the overall population. Thus, the data is insufficient for further multivariate analysis. 

#6 Statistical analysis

The 2nd sentence should be rephrased..

‘Compare the differences in the descriptive characteristics of study participants between 87 Crohn's disease and ulcerative colitis by Chi-square test, the Fisher's exact test (categorical 88 data), the Student’s t-test, or the Mann-Whitney U-test (continuous data), as appropriate.’

Response:

  Thank you for your valuable suggestion. We made the correction in the revised manuscript.

#7 Results

Table 3 should be included in table 2.

Based on changes in the method part include relevant results in table 2 and set table 3 for the results of multivariate analysis

Response:

  Thank you for your valuable suggestion. Multivariate analysis was not performed due to the small case number in the present study.

#8 Conclusion

Please correct  cosmography to tomography.

Response:

  Thank you for your valuable suggestion. We made the correction in the revised manuscript.

Thank you for the opportunity to resubmit this manuscript for consideration for publication in the Medicine If you have any questions or comments regarding this manuscript, please do not hesitate to contact us using the details provided below.

Sincerely,

Hsu-Heng Yen, M.D  

Division of Gastroenterology, Department of Internal Medicine, Changhua Christian Hospital, Changhua, Taiwan

Round 2

Reviewer 3 Report

Authors have tried to improve the manuscript but it still has a lot of grammatical errors and needs editing.

Furthermore the description of the methodology needs also improvement eg you should define which are the exclusion criteria and in the results part you will describe that 3 patients were excluded and why.

There are also many missing information about the population study that would be interesting to check if there is any correlation with CED.

I suggest the authors to include other factors related with disease severity or activity in their analysis, so as to possibly identify more interesting associations and make theirmanuscript more attractive to the reader.

Author Response

Dear Reviewer,

Thank you for reviewing our manuscript and providing your editorial comments as well as reviewers comments for improving our manuscript. Based on these comments, we have made several revisions to our manuscript, which we are hereby resubmitting for your consideration. Our point-by-point responses to the comments are detailed below.

Reviewer’s Suggestion

Authors have tried to improve the manuscript but it still has a lot of grammatical errors and needs editing. Furthermore the description of the methodology needs also improvement eg you should define which are the exclusion criteria and in the results part you will describe that 3 patients were excluded and why.

Response: Thank you for your opinion. We explain why the 3 patients were excluded who had a diagnosis of malignancy before the diagnosis IBD. The diagnosis of malignancy may influence the dose of radiation exposure and therefore, we exclude these three patients for analysis.

There are also many missing information about the population study that would be interesting to check if there is any correlation with CED.I suggest the authors to include other factors related with disease severity or activity in their analysis, so as to possibly identify more interesting associations and make their manuscript more attractive to the readers

Response: Thank you for your opinion. We add the status of surgery and hospital admission of the study population and analyze their relationship to radiation exposure. The finding of IBD-related admission or surgery is associated with higher radiation exposure, which implies the importance of disease control is also helpful to avoid excessive radiation in these patients.

Thank you for the opportunity to resubmit this manuscript for consideration for publication in the Journal of clinical medicine if you have any questions or comments regarding this manuscript, please do not hesitate to contact us using the details provided below.

Sincerely,

Hsu-Heng Yen, M.D  

Division of Gastroenterology, Department of Internal Medicine, Changhua Christian Hospital, Changhua, Taiwan

Fax: +886-4-7228289

Tel: +886-4-7238595ext5501
